# Major Stressful Life Events and the Risk of Pancreatic, Head and Neck Cancers: A Case–Control Study

**DOI:** 10.3390/cancers16020451

**Published:** 2024-01-20

**Authors:** Arthi Sridhar, Vishaldeep Kaur Sekhon, Chandler Nguyen, Kamelah Abushalha, Amirali Tahanan, Mohammad Hossein Rahbar, Syed Hasan Jafri

**Affiliations:** 1Division of Hematology/Oncology, McGovern Medical School, The University of Texas Health Science Center at Houston, Houston, TX 77030, USA; 2Division of Geriatric Medicine and Gerontology, School of Medicine, Johns Hopkins University, Baltimore, MD 21205, USA; 3McGovern Medical School, The University of Texas Health Science Center at Houston, Houston, TX 77030, USA; chandler.h.nguyen@uth.tmc.edu; 4MetroWest Medical Center/Tufts University School of Medicine, Framingham, MA 01702, USA; 5Center for Clinical and Translational Sciences, The University of Texas Health Science Center at Houston, Houston, TX 77030, USA; amirali.tahanan@uth.tmc.edu (A.T.); mohammad.h.rahbar@uth.tmc.edu (M.H.R.)

**Keywords:** Holmes–Rahe, social support, anoikis, prevention, immune suppression

## Abstract

**Simple Summary:**

Several studies have demonstrated the association between stress and cancer. This study investigates the association between major stressful life events and the incidence of head and neck or pancreatic cancer (HNPC). Conducted as a matched case–control study with 280 participants (79 cases, 201 controls), the research utilized the modified Holmes–Rahe stress scale to gather information on major stressful life events. After controlling for variables like sex, age, race, education, marital status, and smoking history, the study demonstrated that patients with HNPC were significantly more likely to report a major stressful life event within the preceding 5 years compared to controls. This highlights a potential link between recent stressful life events and the development of head, neck, and pancreatic cancers, emphasizing the importance of recognizing such events as potential risk factors for malignancies.

**Abstract:**

Background: Major stressful life events have been shown to be associated with an increased risk of lung cancer, breast cancer and the development of various chronic illnesses. The stress response generated by our body results in a variety of physiological and metabolic changes which can affect the immune system and have been shown to be associated with tumor progression. In this study, we aim to determine if major stressful life events are associated with the incidence of head and neck or pancreatic cancer (HNPC). Methods: This is a matched case–control study. Cases (CAs) were HNPC patients diagnosed within the previous 12 months. Controls (COs) were patients without a prior history of malignancy. Basic demographic data information on major stressful life events was collected using the modified Holmes–Rahe stress scale. A total sample of 280 was needed (79 cases, 201 controls) to achieve at least 80% power to detect odds ratios (ORs) of 2.00 or higher at the 5% level of significance. Results: From 1 January 2018 to 31 August 2021, 280 patients were enrolled (CA = 79, CO = 201) in this study. In a multivariable logistic regression analysis after controlling for potential confounding variables (including sex, age, race, education, marital status, smoking history), there was no difference between the lifetime prevalence of major stressful event in cases and controls. However, patients with HNPC were significantly more likely to report a major stressful life event within the preceding 5 years when compared to COs (*p* = 0.01, OR = 2.32, 95% CI, 1.18–4.54). Conclusions: Patients with head, neck and pancreatic cancers are significantly associated with having a major stressful life event within 5 years of their diagnosis. This study highlights the potential need to recognize stressful life events as risk factors for developing malignancies.

## 1. Introduction

The idea that stressful life situations can be linked with the development of various diseases has been known for decades. As early as the 1950s, it was proposed that, in a significant proportion of patients with tuberculosis, a life-organization stress situation of significant proportion appeared shortly before the onset of disease [1]. Since then, stressful life events have been found to be associated with numerous chronic illnesses such as cardiovascular disease, depression, obesity, diabetes, stroke and metabolic syndrome [2,3,4,5,6,7,8]. Psychological stress has also been linked with the aging process in several studies [9,10]. Studies have demonstrated that psychological stress can accelerate the aging process by mechanisms such as telomere shortening; mitochondrial, cellular senescence; chronic inflammation; and stem cell exhaustion [11,12,13,14].

On this note, studies have also been conducted to evaluate the effect of stress on the incidence of malignancies, and varying outcomes have been noted. A prospective study conducted in Denmark did not show any association between stressful life events and the incidence of malignancy [15]. In contrast, a meta-analysis of 165 studies showed an association between stressful life experiences and the incidence of cancer in an otherwise healthy population [16].

The pathophysiology behind this phenomenon has been described in various studies. Cells acquire oncogenic mutations which transform them into malignant cells possessing characteristics known as the “hallmarks of cancer”. Studies have shown that increased levels of circulating catecholamines and glucocorticoids, which are often seen in blood samples from persons undergoing stressful situations, play a role in tumorigenesis via various mechanisms, including angiogenesis, resistance to anoikis, metastasis and increased cell survival [17,18]. In vivo studies have hypothesized that that chronic stress affects tumor angiogenesis by the chronic release of proangiogenic factors (e.g., VEGF, IL-6, TGF-α and -β and TNF-α) and causes immune suppression, which can increase the risk of cancer development [19,20].

Social support has been shown in several studies to play a role in cancer progression [21]. The effect of social support on cancer progression is still debated and the evidence is variable. Some studies suggest that social support can have an impact on cancer progression. Several studies in cancer include social support in some form, and further research is still needed in this area [16,22,23].

Head and neck cancers account for more than 600,000 new cases and around 300,000 deaths annually [24]. Pancreatic cancers accounted for approximately 495,773 new cancer diagnoses and 466,003 deaths in 2020 [25]. Head, neck, and pancreatic cancers are associated with significant morbidities after diagnosis and are associated with a myriad of challenges for patients and their caregivers [17]. It is becoming more evident that with the advances in available pharmaceutical interventions for malignancies, it is also important to identify modifiable risk factors and engage in primary prevention. The association between stress and cancers has been shown in other studies, particularly in colon, breast and lung cancer [26,27,28]. In this study, we report our findings on the association between major stressful life events and the incidence of head and neck or pancreatic cancer (HNPC).

## 2. Methods

This matched case–control study was conducted at The University of Texas Health Science Center (UTHealth), Memorial Hermann Cancer Center and Memorial Hermann Hospital Texas Medical Center and was approved by the institutional review board of UTHealth. Survey data were collected from patients receiving treatment at the above-mentioned institutions from 1 January 2018 to 31 August 2021. Written consent was collected from all the patients enrolled in the study.

The inclusion criteria included cases (CAs) who were patients diagnosed with HNPC in the preceding 12 months, aged ≥ 40 years who were receiving treatment at the aforementioned institutes during the data collection period. Controls (COs) were patients receiving treatment for any causes other than malignancy at outpatient pulmonary clinics or who were admitted at Memorial Hermann Hospital. Controls were matched with cases by age and smoking status. Patients were excluded from the study if they reported any one of the following events in the preceding 12 months: admission to an intensive care unit (ICU), two or more hospital admissions due to any cause or residing in a nursing home. Patients were also excluded if they had a previous diagnosis of HNPC or if they had a prior history of other malignancies within the past 5 years. Information on major stressful life events was collected using the modified Holmes and Rahe stress scale [29]. Social support was assessed using the Duke-UNC functional social support scale [30].

### 2.1. Patient Recruitment

A total of 283 patients being seen at UT/Memorial Hermann hospitals and clinics without history of a cancer diagnosis were approached for the study, out of which, 220 patients consented to the study and were willing to fill out the Appendix A. Amongst the 220 patients who consented to the study, 19 patients were excluded due to incomplete information for analysis and 201 patients were enrolled as controls. Amongst the patients with HNPC, 170 patients were approached to be a part of the study, and 88 patients consented and were enrolled as cases. Nine patients were excluded due to incomplete medical records for analysis.

### 2.2. Data Analysis

The modified Holmes and Rahe Stress score assessed the occurrence of major stressful events (death of a spouse, death of a child/immediate family member, serious personal illness, divorce/separation, loss of a job, caring for ill family member, financial difficulties, relocation, stress at work, detention/incarceration and retirement). The social support score was calculated using the Duke-UNC functional social support questions.

Descriptive analysis was performed to examine the association between HNPC with all demographics, stress, social support and medical history variables. For categorical variables, χ^2^ or univariate logistic regression analysis was performed. For continuous variables, we used a two-sample *t*-test if the distributions were normal and the Wilcoxon rank-sum test if the distributions were not normal. Inferential analysis was performed using a logistic regression model to assess the relationship between HNPC and stress while controlling for social support, medical history and demographic variables. All data analyses were performed using SAS version 9.4 software (SAS Institute Inc., Cary, NC, USA).

The primary endpoint was the odds of experiencing a major stressful life event in cases versus controls. A sample of 280 patients (CA = 79 and CO = 201) was estimated to be sufficient to provide a statistical power of 80% or more to detect an odds ratio (OR) of 2.00 or higher with a significance level of 5%.

## 3. Results

A total of 280 patients were enrolled in the study (CA = 79, CO = 201). Among cases, 41 patients had a history of pancreatic cancer and 38 had head and neck cancer. Table 1 shows the characteristics of patients.

The mean age of cases was 63 years, and that of controls was 64 years. Most study participants were white (CA = 53%, CO = 46%), followed by African American (CA = 21%, CO = 37%), Asian (CA = 14%, CO = 2%), Hispanic (CA = 7%, CO = 12%), and other (CA = 6%, CO = 2%). There were significantly more men in the case group than in the control group (Ca = 66%, CO = 49%; *p* = 0.013). A higher proportion of cases were married than controls were (CA = 65%, CO = 48%; *p* = 0.015). There was no difference in education status between the two groups. Controls were more likely to report high blood pressure (CA = 57%, CO = 73%; *p* = 0.009), COPD (CA = 9%, CO = 33%; *p* < 0.001) and cardiovascular disease (CA = 16%, CO = 48%; *p* < 0.001). The median numbers of years that cases and controls reported smoking exposure was not significantly different between the groups (CA = 36 years, CO = 37 years). There was no difference in the incidence of major depression between groups.

As shown in Table 2, we evaluated the odds of experiencing major stressful life events in cases and controls. There was no statistically significant difference in the odds of having a major stressful life event over one’s lifetime between cases and controls (CA = 88.6% vs. CO = 93.53%; *p* = 0.173). However, when restricted to the preceding 5 years, cases had 50% higher odds of having a major stressful life event than controls (CA = 74.68% vs. 65.67%; odds ratio (OR) = 1.54; *p* = 0.147).

Cases demonstrated better social support (*p* = 0.018, OR = 1.06, 95% confidence interval (CI), 1.01–1.11). When various interactions were not controlled for, the results were not statistically significant. In the multivariable analysis (Table 3), after controlling for interactions such as age, sex, race, education, marital status, beta blocker usage and social support, patients with HNPC had significantly higher odds of having experienced a major stressful life event within the preceding 5 years than controls (*p* = 0.01, OR = 2.32, 95% CI, 1.18–4.54). Moreover, the use of beta blockers appears to be protective (*p* = 0.0002, OR = 0.29, 95% CI, 0.15–0.56).

## 4. Discussion

Our analysis shows that patients with any stressful life event in the preceding 5 years were more likely to develop HNPC. This was found to be statistically significant after controlling for various interactions. To our knowledge, this is the first study using a matched case–control study to examine the association between stressful life events and a diagnosis of HNPC.

The association of stressful experiences with the development of malignancies has been explored in various studies. It has been demonstrated that there is a perception among cancer patients that the incidence of their cancers may be linked with exposure to stressful situations in their lives [31]. The relationship between stressful life experiences and breast cancer has been well studied, with varying results. A meta-analysis of seven studies demonstrated the association between striking life events and the incidence of breast cancer. Striking life events were defined as an anxiety disorder precipitated by life events including a change in marital status, such as separation, divorce or widowhood; the death of a spouse, child or close relative; a friend’s illness; personal health problems; and a change in financial status. The study showed that women with striking life events had a 1.5-fold greater risk of developing breast cancer (*p* = 0.003, 95% CI, 1.15–1.97). In a prospective case–control study, Jafri et al. used a modified Holmes–Rahe scale to record stressful life events in patients with lung cancer and healthy controls. They demonstrated that patients with newly diagnosed lung cancer were more likely to have had a major stressful life event in the preceding 5 years (OR = 1.78, *p* = 0.03) [28].

The pathophysiology behind the association between stressful events and cancer is attributed to the release of glucocorticoids through the hypothalamic–pituitary axis molecular pathways and in beta-adrenergic signaling through the sympathetic nervous system [28,32]. The release of catecholamines can stimulate the transcription of various proteins involved in the regulation of cellular growth, differentiation and the transcription of various genes, which can result in the onset and progression of various malignancies [33,34,35]. The role of cortisol in tumor cell proliferation has also been described in various malignancies. The role of glucocorticoid-activated receptors in cell growth and survival has been described in various studies [17,33]. For example, glucocorticoid-inducible kinase 3 has been described as a potential oncogene in nasopharyngeal carcinoma [34]. A few studies have shown the impact of circulating catecholamines and glucocorticoid levels in oral cancer: that chronic stress is associated with increased IL-6 expression in humans with oral cancer and increased VEGF expression with tumor size in oral carcinoma mouse models via increased circulating catecholamines and glucocorticoids [18,35]. Similarly, tobacco-specific carcinogens have been found to stimulate beta-adrenergic signaling, resulting in the proliferation of pancreatic duct epithelium in humans [36,37].

The well-known risk factors for pancreatic cancer include alcohol consumption, diabetes mellitus, recurrent pancreatitis, obesity, chemical exposures, racial risk and inherited genetic syndromes. Similarly, the well-known risk factors associated with head and neck cancers include alcohol and tobacco use, HPV infections, occupational exposures, radiation exposures, Epstein–Barr virus infections and racial and genetic predispositions [38]. Although it is difficult to control for all possible risk factors, in our study, we were able to control for certain confounding factors such as age and race. Our study also demonstrated that patients receiving beta blockers were less likely to have HNPC when compared to those not receiving these medications. This is consistent with other studies describing the protective effects of beta blockers due their antiangiogenic effects and their effect on the tumor microenvironment [39,40].

HNPC are malignancies associated with significant morbidity and reductions in quality of life even in the curative setting. We believe that the primary prevention of HNPC with the involvement of mental health rehabilitation should be considered in at-risk individuals. With the available data and research, it is evident that stress influences the tumor microenvironment and immune suppression, which can precipitate the incidence of cancers in patients. It is possible that in the presence of existing risk factors for cancer development such as smoking and obesity, one or more major stressful event can lead to the activation of sympathetic pathways and immune suppression and thus precipitates sufficient physiological changes resulting in cancer development. However, this hypothesis requires further research to correlate the tumor microenvironment with the onset of major stressful events, which may be challenging in the prospective setting.

The term “major stressful life events” is self-explanatory, referring to events that can significantly impact individuals. Through our study, we identified a clinical correlation between these events and HNPC. This presents an opportunity to enhance our social structure by integrating mental health and rehabilitation into primary care. By doing so, our primary healthcare system can play a vital role in raising awareness about self-help tools and resources, aiding individuals facing life-altering events in coping with the associated changes. This approach advocates for a holistic perspective on healthcare, addressing both physical and psychological well-being, and emphasizes the importance of early intervention and preventive measures to support individuals at risk of developing mental health issues during such events.

## 5. Limitations

The main limitation of the study is its small sample size, making it difficult to generalize the results. Our second limitation is recall bias, as this study was based on self-reported surveys and had questions related to past events. Furthermore, the development of cancer is a complex process that involves various epidemiological and genetic factors. This study included both head and neck and pancreatic cancer patients. There are differences between these two cancers with respect to aggressiveness and other epidemiological and etiological factors. For example, pancreatic cancers have a high fatality rate and have etiological factors different from those of head and neck cancers [41]. This study did not consider the genetic, hereditary and family history of these patients, which may also influence the incidence of HNPC cancers in the study population. This study also did not control for other variables like sex, race, marital status, and chronic illnesses like hypertension, COPD and cerebrovascular diseases. However, it is worthwhile to note some of the variables associated with malignancies were found to be higher in controls when compared to cases. This gives more credence to our observation that despite having more chronic illness amongst controls, major stressful life events likely contributed significantly to the development of cancer amongst cases.

The Holmes–Rahe scale has been validated in several studies and used extensively to study the impact of psychological stress. It has been a valuable tool in understanding the link between life events and stress, but it has several limitations that we should be aware of. The tool assigns a fixed-point value to each event, not assessing the impact of individual differences in coping mechanisms, resilience and subjective perception of stress. The original research conducted for the HRSS dates back to the 1960s, during which, the original study participants were primarily married, middle-aged and of a specific socioeconomic class, limiting the generalizability of findings to other populations. With evolving time, the tool likely needs re-evaluation.

## 6. Conclusions

In conclusion, our study demonstrates an association between major stressful life events and the incidence of head and neck and pancreatic cancers in subjects who experiences these events in the 5 years preceding the diagnosis of their malignancy. There are numerous studies that have been conducted that have shown an association between psychological stress and chronic illness, accelerated ageing and various malignancies. It is becoming increasingly important to address psychological stress at institutional and national levels. It may be valuable to replicate our study using a larger sample size with one type of cancer at a time and to include patients’ biochemical, viral and biomarker history.

## Figures and Tables

**Table 1 cancers-16-00451-t001:** Patient characteristics for cases (with head and neck or pancreatic cancer) and controls based on univariable logistic regression model (N = 280 participants).

Variable	Cases (N = 79)	Controls (N = 201)	OR (95% CI)	*p* Value *
Mean (SD) age, years	62.68 (10.16)	64.38 (10.54)	0.98 (0.96–1.01)	0.223 **
Sex				
Male	52 (65.82)	99 (49.25)	1.98 (1.16–3.41)	0.013
Female	27 (34.18)	102 (50.75)	0.50 (0.29–0.87)	
Race				
White	38 (52.78)	93 (46.27)	1.08 (0.64–1.81)	0.782
African American	15 (20.83)	74 (36.82)	0.40 (0.21–0.76)	0.005
Hispanic	5 (6.94)	24 (11.94)	0.50 (0.18–1.36)	0.173
Asian	10 (13.89)	5 (2.49)	5.68 (1.88–17.20)	0.002
Other	4 (5.56)	5 (2.49)	2.09 (0.55–8.00)	0.281
Marital status				
Married	51 (64.56)	97 (48.26)	1.95 (1.14–3.34)	0.015
Single	11 (13.92)	31 (15.42)	0.89 (0.42–1.87)	0.752
Separated	3 (3.80)	15 (7.46)	0.49 (0.14–1.74)	0.270
Divorced	11 (13.92)	28 (13.93)	1.00 (0.47–2.12)	0.999
Widowed	3 (3.80)	30 (14.93)	0.23 (0.07–0.76)	0.016
Maximum education				
Some school	10 (12.66)	20 (10.10)	1.31 (0.59–2.94)	0.511
High school	29 (36.71)	80 (40.40)	0.88 (0.51–1.50)	0.633
Some college	17 (21.52)	55 (27.78)	0.73 (0.39–1.35)	0.315
College graduate	16 (20.25)	31 (15.66)	1.39 (0.71–2.72)	0.332
Masters	6 (7.59)	7 (3.54)	2.28 (0.74–7.01)	0.151
PhD	1 (1.27)	5 (2.53)	0.50 (0.06–4.37)	0.533
High blood pressure				
Yes	45 (56.96)	147 (73.13)	0.49 (0.28–0.84)	0.009
No	34 (43.04)	54 (26.87)	2.06 (1.19–3.54)	
COPD				
Yes	7 (8.86)	65 (32.66)	0.20 (0.09–0.46)	<0.001
No	72 (91.14)	134 (67.34)	4.99 (2.17–11.45)	
Inflammatory arthritis				
Yes	5 (6.33)	25 (12.50)	0.47 (0.17–1.28)	0.141
No	74 (93.67)	175 (87.50)	2.11 (0.78–5.74)	
Cardiovascular disease				
Yes	13 (16.46)	95 (47.50)	0.22 (0.11–0.42)	<0.001
No	66 (83.54)	105 (52.50)	4.59 (2.38–8.85)	
Major depression				
Yes	11 (13.92)	30 (15.00)	0.92 (0.44–1.93)	0.819
No	68 (86.06)	170 (85.00)	1.09 (0.52–2.30)	
Use of beta blockers				
Yes	22 (28)	99 (49)	0.43 (0.24–0.76)	0.003
No	53 (67)	102 (51)	2.34 (1.32–4.13)	
Smoking exposure years, median (IQR)	35 (27, 41)	37.5 (30, 44)	0.98 (0.94, 1.02)	0.275 **

Note: CI, confidence interval; COPD, chronic obstructive pulmonary disease; OR, odds ratio. Data are no. (%) unless otherwise indicated. * *p*-values for categorical variables are calculated based on logistic regression. ** *p*-values are calculated based on two-sample *t*-test for normally distributed continuous variables and Wilcoxon rank-sum test when the distributions were not normal.

**Table 2 cancers-16-00451-t002:** Major stressful life events among participants in the past (using the modified Holmes–Rahe stress scale).

Variable	Cases (N = 79)	Control (N = 201)	Odds Ratio (95% CI)	*p* Value
Any stressful event ever in lifetime	70 (88.60%)	188 (93.53%)	0.54 (0.22–1.13)	0.173
Social support score, median (IQR)	40 (33, 40)	37 (31, 40)	1.06 (1.01–1.11)	0.018
Any stressful life events in the past 5 years				
Yes	59 (74.68%)	132 (65.67)	1.54 (0.86–2.77)	0.147
No	20 (25.32)	69 (34.33)	0.65 (0.36–1.16)	

Note. Data are no. (%) unless otherwise indicated. CI, confidence interval; IQR, interquartile range; OR, odds ratio.

**Table 3 cancers-16-00451-t003:** Factors associated with head, neck or pancreatic cancer based on multivariable logistic regression in an additive model.

Variable	Adjusted OR (95% CI)	*p*-Value
Sex (male vs. female)	1.84 (1.01–3.37)	0.05
Age	0.99 (0.96–1.02)	0.57
Race (white vs. others)	0.92 (0.50–1.68)	0.78
Maximum education (high school vs. others)	0.81 (0.45–1.49)	0.50
Marital status (married vs. others)	2.64 (1.36–5.14)	0.004
Any stressful life events in preceding past 5 years (yes vs. no)	2.32 (1.18–4.54)	0.01
Social support score (≥35 vs. <35)	1.76 (0.91–3.39)	0.09
Beta blocker use (yes vs. no)	0.29 (0.15–0.56)	0.0002

## Data Availability

The datasets analyzed for this study will be made available upon request to the corresponding author.

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
