# Peer review of "Major Stressful Life Events and the Risk of Pancreatic, Head and Neck Cancers: A Case–Control Study"

_cancers, 2024, doi:10.3390/cancers16020451_

Round 1
Reviewer 1 Report
Comments and Suggestions for Authors
Sridhar et al present an interesting study about the association between stressful life and subsequent head, neck, and pancreatic cancers. Patients with head, neck and pancreatic cancers are significantly associated with having a major stressful life event within 5 years of their diagnosis.
I have the following comments:
1. Authors should clearly define the study goal in the abstract (background)
2. Authors should better explain why they select defined diagnoses and therapies to adjust for (like COPD, beta blockers and on) since they did not include important other diagnoses which are known to be risk factors for cancer (diabetes). The lack of some diagnoses is a very large study limitation.
3. “The modified Holmes and Rahe Stress score assessed the occurrence of major stressful events (death of a spouse, death of a child/immediate family member, serious personal illness, divorce/separation, loss of a job, caring for ill family member, financial difficulties, relocation, stress at work, detention/incarceration, and retirement).” I see here a big methodological issue. When cancer patients are allowed to have cancer in last five years (as per study definition) when cancer patients can list cancer as one of the events in their life. This cause automatically the strong relationship between cancer in the past (last 12 months) as event and cancer as outcome.
4. Patients in the cancer cases were less often widowed and separated than controls (Table 1) what may either mean that these events were more than 5 years prior to index date, or other events were much more often in cases. This should be discussed.
Author Response
Dear Reviewer,
Thank you very much for your response and taking the time to review our manuscript. Please see below our responses to your comments.
- Authors should clearly define the study goal in the abstract (background)
- Answer: Thank you for your input, we had included the study goal in the methods but have moved it to the background per your request.
- Authors should better explain why they select defined diagnoses and therapies to adjust for (like COPD, beta blockers and on) since they did not include important other diagnoses which are known to be risk factors for cancer (diabetes). The lack of some diagnoses is a very large study limitation.
- Answer: We chose to control for interactions specifically age, sex, race, education, marital status, beta-blocker usage, and social support. While there are numerous factors that can influence the impact of stress on human physiology, we chose to adjust for these interactions since there is significant data to support the impact of these demographics on individual stress. Beta-blocker use has been demonstrated in several studies to reduce the adrenergic response to stress. This has also been seen in prior papers published by the corresponding author. For these reasons, a decision was made to also control for betablocker use.
- References:
- Jafri SH, Ali F, Mollaeian A, et al. Major Stressful Life Events and Risk of Developing Lung Cancer: A Case-Control Study. Clinical Medicine Insights: Oncology. 2019;13. doi:10.1177/1179554919835798
- Peixoto R, Pereira ML, Oliveira M. Beta-Blockers and Cancer: Where Are We? Pharmaceuticals (Basel). 2020 May 26;13(6):105. doi: 10.3390/ph13060105. PMID: 32466499; PMCID: PMC7345088.
- “The modified Holmes and Rahe Stress score assessed the occurrence of major stressful events (death of a spouse, death of a child/immediate family member, serious personal illness, divorce/separation, loss of a job, caring for ill family member, financial difficulties, relocation, stress at work, detention/incarceration, and retirement).” I see here a big methodological issue. When cancer patients are allowed to have cancer in last five years (as per study definition) when cancer patients can list cancer as one of the events in their life. This cause automatically the strong relationship between cancer in the past (last 12 months) as event and cancer as outcome.
- Answer: Patients were excluded from the study if they had a previous diagnosis of HNPC or if they had a prior history of other malignancies within the past 5 years. Questionnaires included clear instructions in answering the questions and serious personal illness included any illness other than their current diagnosis of head and neck or pancreatic cancer. With our careful screening and the clear nature of the questionnaires, we did our best to address this concern. A direct quote from the questionnaire- “Do you have a history of serious personal illness other than current cancer”, demonstrates that we excluded current cancer as major event in the study.
- Patients in the cancer cases were less often widowed and separated than controls (Table 1) what may either mean that these events were more than 5 years prior to index date, or other events were much more often in cases. This should be discussed.
- Answer: In our study, controls were more commonly found to be widowed or separated. However, in our multivariate analysis, we controlled for marital status as one of the variables during data analysis.
Reviewer 2 Report
Comments and Suggestions for Authors
Dear Authors,
Thank you very much for the opportunity to review this manuscript. The results address a highly important public health issue. The high level of social stress, which has reached very high values in recent years (e.g. the SARS-CoV-2 virus pandemic), may have its biological consequences in the near future.
Comments and doubts about the manuscript:
Notes on methodology:
1. Please describe in detail (including source materials) which version of the Holmes and Rahe scale was used? What was the modification of this scale? Why do the authors not refer to the original starting scale)? Please make appropriate additions to the text of the manuscript.
2. Please explain cutt-point on the social support scale? Please provide relevant literature.
3. Figure 1 duplicates the content from Table 2 - this is not advisable. In my opinion, Figure 1 is unnecessary.
4. The study and control groups differ significantly in gender distribution, race, marital status and other characteristics - including the important blood pressure (Table 1). This may significantly affect the results. It has been proven that respondents with different marital status differ significantly in their biological condition, e.g. blood pressure (https://pubmed.ncbi.nlm.nih.gov/16080590/). Marital status (https://pubmed.ncbi.nlm.nih.gov/35639382/). The authors do not address these issues, but in my opinion they should.
Please refer to it in the limitation study section. Additionally, as Table 3 shows, marital status turns out to have a stronger relationship than stress with the risk of developing the analyzed cancers. All the more reason, the authors should focus more attention on this feature, also in the theoretical sections of the manuscript.
5. As can be seen directly from Table 2, there are no significant differences in the level of total stress or 5-year stress between the research and control groups. Both p values are non-significant and the Odds Ratio values are identical. Why do the authors focus their thinking on 5-year stress and not on the overall stress? Please explain.
Notes on theoretical chapters:
1. The topic is very interesting and broad. In my opinion, the authors should demonstrate in the Introduction that stress (studied with an analogous tool) is more strongly related to other biological features or biological processes than other features or behaviors. Research results prove that regardless of lifestyle, stress accelerates the aging process (https://www.mdpi.com/1660-4601/19/9/5044/xml) and affects BMI or body fat content (https://www.mdpi.com/1660-4601/19/19/12149). These aspects were omitted by the authors.
2. The scale used to measure stress has numerous limitations. Please refer to this issue in the Limitation Study.
3. The Limitation Study section is missing from the manuscript. Please complete it.
Kind regards,
reviewer
Author Response
Dear Reviewer,
Thank you for your comments. We appreciate your time in reviewing this manuscript. Please see our responses below:
Notes on methodology:
- Please describe in detail (including source materials) which version of the Holmes and Rahe scale was used? What was the modification of this scale? Why do the authors not refer to the original starting scale)? Please make appropriate additions to the text of the manuscript.
- Answer: The Holmes and Rahe Stress scale has 43 stressful events. We focused our study on Major stressful events (having a high score on Holmes and Rahe stress scale) only. Thus, we modified the H-R stress scale by restricting it to include 11 stressful events only. (See attached questionnaire).
- Please explain cutt-point on the social support scale? Please provide relevant literature.
- Answer: We used the mean of the social support score (mean=35) across all participants in both the cases and control groups as the cutoff point for dichotomizing this variable in the statistical analysis. The reference is in the manuscript, and we have included it below for your reference. This scale has been used in numerous studies related to social support in patients with cancers. We have included some of the references here.
- References:
- Broadhead WE, Gehlbach SH, de Gruy FV, Kaplan BH. The Duke-UNC Functional Social Support Questionnaire. Measurement of social support in family medicine patients. Med Care. 1988 Jul;26(7):709-23. doi: 10.1097/00005650-198807000-00006. PMID: 3393031.
- Gonzalez-Saenz de Tejada M, Bilbao A, Baré M, Briones E, Sarasqueta C, Quintana JM, Escobar A; CARESS-CCR Group. Association between social support, functional status, and change in health-related quality of life and changes in anxiety and depression in colorectal cancer patients. Psychooncology. 2017 Sep;26(9):1263-1269. doi: 10.1002/pon.4303. Epub 2016 Dec 19. PMID: 28872742.
- Martín-Abreu CM, Hernández R, Cruz-Castellanos P, Fernández-Montes A, Lorente-Estellés D, López-Ceballos H, Ostios-Garcia L, Antoñanzas M, Jiménez-Fonseca P, García-García T, Calderon C. Dignity and psychosocial related variables in elderly advanced cancer patients. BMC Geriatr. 2022 Sep 5;22(1):732. doi: 10.1186/s12877-022-03423-7. PMID: 36064353; PMCID: PMC9446795.
- Figure 1 duplicates the content from Table 2 - this is not advisable. In my opinion, Figure 1 is unnecessary.
- Answer: Thank you for this comment, we agree and will remove figure 1.
- The study and control groups differ significantly in gender distribution, race, marital status and other characteristics - including the important blood pressure (Table 1). This may significantly affect the results. It has been proven that respondents with different marital status differ significantly in their biological condition, e.g. blood pressure (https://pubmed.ncbi.nlm.nih.gov/16080590/). Marital status (https://pubmed.ncbi.nlm.nih.gov/35639382/). The authors do not address these issues, but in my opinion they should.
- Answer: The two groups were controlled for age and smoking status. The groups were not controlled for many other variables like sex, race, marital status, and chronic illnesses like hypertension, COPD and CVD. Controls were significantly more likely to have chronic illness like HTN, COPD and CVD as compared to cases (see table 1) which is actually linked with higher incidence of cancer (See Reference: Tu H, Wen C P, Tsai S P, Chow W, Wen C, Ye Y et al. Cancer risk associated with chronic diseases and disease markers: prospective cohort study BMJ 2018; 360 :k134 doi:10.1136/bmj.k134). This gives more credence to our observation that despite having more chronic illness amongst controls, major stressful life events likely contributed significantly to development of cancer amongst cases.
- Please refer to it in the limitation study section. Additionally, as Table 3 shows, marital status turns out to have a stronger relationship than stress with the risk of developing the analyzed cancers. All the more reason, the authors should focus more attention on this feature, also in the theoretical sections of the manuscript.
- Answer: Marital status is linked with improved cancer specific mortality in population-based study. In our study controls were significantly more likely to be married again giving more credence to our observation that major stressful life events amongst cases were important contributor to development of cancer. (See Reference: Zhu, S., Lei, C. Association between marital status and all-cause mortality of patients with metastatic breast cancer: a population-based study. Sci Rep 13, 9067 (2023). https://doi.org/10.1038/s41598-023-36139-8). We will include a section on limitations in our manuscript
- As can be seen directly from Table 2, there are no significant differences in the level of total stress or 5-year stress between the research and control groups. Both p values are non-significant and the Odds Ratio values are identical. Why do the authors focus their thinking on 5-year stress and not on the overall stress? Please explain.
- Answer: Table 2 highlights the findings in a univariate analysis. However, when we control for various factors as highlighted in the manuscript, a significant relationship is found between the incidence of stressful life events in the past 5 years and cancers. We focused on stressful life events within the preceding five years because we hypothesize that Major stressful life events is more likely have an impact on cancer development soon after the event as opposed to other risk factors for cancer like smoking which can take decades to show its impact. Moreover, in our clinical and prior observations (See Ref) major stressful events are more likely to have impact in soon after the event. As can be seen from the data that as compared to life time stressful events which were higher amongst controls, the incidence of majors stressful events increased amongst cases though not statistically significant.
- Reference:
- Jafri SH, Ali F, Mollaeian A, et al. Major Stressful Life Events and Risk of Developing Lung Cancer: A Case-Control Study. Clinical Medicine Insights: Oncology. 2019;13. doi:10.1177/1179554919835798
Notes on theoretical chapters:
- The topic is very interesting and broad. In my opinion, the authors should demonstrate in the Introduction that stress (studied with an analogous tool) is more strongly related to other biological features or biological processes than other features or behaviors. Research results prove that regardless of lifestyle, stress accelerates the aging process (https://www.mdpi.com/1660-4601/19/9/5044/xml) and affects BMI or body fat content (https://www.mdpi.com/1660-4601/19/19/12149). These aspects were omitted by the authors.
- Answer: Thank you for you input. We have included a section on this topic in our introduction.
- The scale used to measure stress has numerous limitations. Please refer to this issue in the Limitation Study.
- Answer: The Holmes-Rahe stress scale has been standard tool to assess major stressful events and is a verified tool. We modified to focus on major stressful events only and limited it to 11 major stressful events as opposed to 43 mentioned in the original scale. However, there are limitations to this measurement tool which we will incorporate in our manuscrtip.
- The Limitation Study section is missing from the manuscript. Please complete it.
- Answer: We have included a section on limitations in the manuscript.
Round 2
Reviewer 1 Report
Comments and Suggestions for Authors
ok
Reviewer 2 Report
Comments and Suggestions for Authors
Dear Authors,
Thank you for the opportunity to review this paper again.
The text of the publication has been improved.
Thank you very much for this additions. This is exactly what should have been added to the manuscript.
I am satisfied with the revised version of the manuscript.
Thank you for the opportunity to review this article.